# Study on the Passive Heating System of a Heated Cooking Wall in Dwellings: A Case Study of Traditional Dwellings in Southern Shaanxi, China

**DOI:** 10.3390/ijerph18073745

**Published:** 2021-04-03

**Authors:** Simin Yang, Bart J. Dewancker, Shuo Chen

**Affiliations:** Faculty of Environmental Engineering, The University of Kitakyushu, Kitakyushu 808-0135, Japan; bart@kitakyu-u.ac.jp (B.J.D.); chensure8718@163.com (S.C.)

**Keywords:** passive heating system, cooking heated wall, dwellings, southern Shaanxi, simulation analysis

## Abstract

In China, research on winter heating and energy saving for residential buildings mainly focuses on urban residences rather than rural ones. According to the 2018 China Building Energy Consumption Research Report, rural residential buildings emit about 423 million tons of carbon, accounting for 21% of the country’s total carbon emissions. According to the research on China’s greenhouse gas inventory, the main sources of carbon emissions in rural areas are from cooking and the burning of fuelwood and biomass for heating in winter. In this study, the southern Shaanxi area, which is hot in summer and cold in winter, was selected as the research site, and a fire wall system was planned that combines cooking and heating facilities in residential buildings. The system uses the heat generated by cooking and the heat storage capacity of the wall, as well as the principle of thermal radiation and heat convection, to increase the indoor temperature. The advantage is that the hot air generated is mainly concentrated in the inside of the wall, which reduces the direct contact with the cold outdoor air and avoids excess heat loss. In this study, in addition to considering the influence of the cooking fire wall system on the indoor temperature, the difference in the outer wall with or without solar thermal radiation was also considered. The research results show that the use of a cooking fire wall heating system reduces the annual heat load of the building to 440.8318 KW·h, which is a reduction rate of 7.91%. When there is solar radiation on the outer wall, the annual thermal load of the building is reduced by 1104.723 kW·h, and the reduction rate is 19.84%.

## 1. Introduction

### 1.1. Motivation

Nowadays, with the vigorous development of construction, industry, transportation, agriculture, and other industries, total global energy consumption has constantly increased [1]. The ratio of energy consumption in the building industry was 29.36% [2], making the industry one of the main areas of energy consumption in 2018 [3]. Based on the current energy consumption of the building industry, it is estimated that the carbon emissions of the building industry will increase to 50% in 2050 [4]. For developing countries, the energy consumption of the construction industry is even more severe [5]. The building industry accounts for 46.5% of the total energy consumption and 51.3% of the national carbon emissions in China [6]. However, China’s building industry for energy demand and carbon emissions will continue to increase [7]. It is expected that in 2050, these numbers will triple [8]. The residential sector is the main component of the building industry [9]. The residential energy consumption increased from 291,435 to 346,713 Ktoe from 1990 to 2018 [2]. Thus, the residential energy consumption has shown a rapid increase trend. In response to this trend phenomenon, many experts and scholars have proposed green, zero-energy, low-carbon, and other types of residential buildings to reduce the demand for energy [10,11].

To reduce the energy demand of residential buildings, national government departments have also put forward urban residential energy conservation design codes and related laws and regulations in Chian [12]. Many experts and scholars have put forward many research strategies for urban residential energy conservation through field research, data collection, model analysis, mathematical calculations, and other research methods [13]. However, due to the diversity of residential buildings in rural areas of China and the fact that most of them are self-built, residents pay little attention to energy conservation during construction [14]. As of 2018, the total rural residential construction area is about 23.8 billion m^2^ [6]. Hence, it is necessary to design energy-saving dwellings in rural areas. However, there are essential differences between urban residences and rural residences, such as building form, height, material, space function, etc., so it is not feasible to simply apply the energy-saving designs used for urban residences to rural residences [15]. Compared with urban residential buildings, rural residential buildings have advantages of being able to be more flexible in terms of space and having larger sites. Therefore, it is convenient to implement the construction of rural residential buildings [16]. In this regard, when designing passive energy-saving technologies for residential buildings, the local living habits and natural conditions should be fully considered, and energy-saving design concepts should be applied to rural residential buildings [17]. This is of great significance to the design of modern rural residences [18]. This research’s main proposal is to establish a new passive cooking and heating system in winter, effectively using production and lifestyle to increase indoor temperature and reduce energy consumption.

### 1.2. Literature Studies

Research on passive heating design is mainly divided into three categories: The first is climate adaptation strategy research [19], mainly including reasonable orientation [20], the optimization of the envelope structure [21], appropriate window-to-wall ratio (WWR), and so on [22]. The second is the use of renewable energy in buildings, such as solar energy and geothermal energy [23]. The third is the study of Kang [24].

The research on building orientation mainly focuses on studying the relationship between different orientations, indoor temperature, and energy savings to find the most reasonable orientation for a building [25]. The research results of Wang et al. [26] show that in cold areas, the orientation of buildings is a factor in obtaining more solar radiation. Torreggiani et al. [27] showed that direction had an effect on indoor temperature through simulation calculation. Chi et al. [28] rotated traditional residential buildings clockwise by 20°. Through simulation calculation and comparisons of temperature, humidity, and lighting in 18 sets of data, the research results showed that the best angle was S and the worst angle was N-W80 for the whole year, and the difference in building electricity consumption between the two was 150 kWh. The research results show that a good orientation has the potential to save energy. The optimization research on envelope structures mainly focuses on improving the thermal performance of the wall structure, thereby improving the thermal comfort of human body and reducing carbon emissions. Liu et al. [29] conducted research on traditional dwellings and cave dwellings in northwest China, and their results showed that thick walls can effectively restrain changes in indoor temperature and keep the internal surface temperature of a wall stable. This research provides a scientific basis for reducing energy demand. Xu et al. [30] measured indoor temperature, humidity, and lighting data of brick and earth walls in winter and summer through field measurements of traditional dwellings with two types of walls in Qinba mountain. The results showed that dwellings with earth walls were more suitable for the local climate. For brick walls, they recommended using a variety of materials such as concrete, porous brick, and foam concrete to achieve a good thermal insulation performance and low cost. This study provides useful data for future research on folk dwellings. Research on appropriate WWRs mainly focuses on the influence of WWR on indoor temperature. Chi et al. [31] divided the WWR of traditional houses in Zhejiang into categories of 0.1–0.9 and calculated the indoor lighting, temperature, humidity, and wind speed through software simulation. The range of the three intersecting values is the best WWR. Obrecht et al. [32] studied the glazing ratio of glass facing different directions and concluded that using different glazing ratios in different areas can meet different cooling and heating requirements.

Through the energy-saving design of residential buildings, buildings can use renewable energy to improve their indoor temperatures, such as solar or geothermal energy [33]. Research on passive solar technology mainly comprises two parts: the influence of solar technology on indoor temperature and measures for the improvement of solar technology. Passive solar technology generally uses a combination of different types of glass and buildings to effectively increase indoor temperature, such as the roof of a greenhouse [34], the Trombe wall [35], and the solar chimney [36]. Cossu et al. [37] stated that the roof of a greenhouse mainly refers to the use of a glass device covering a roof to increase the solar radiation and contribute to indoor heat, thus affecting indoor temperature. Trombe walls are another passive solar technology [38]. From the perspective of reducing the dependence on energy and carbon emissions, Irshad et al. [39] proved that double-layer glass with argon gas can effectively improve indoor temperature when placed on a Trombe wall. By comparing the thermal performance of traditional and composite Trombe walls, Chen et al. [40] used the ANSYS software to simulate the calculation, and the research results showed that composite Trombe walls had a better thermal performance. Solar chimneys usually protrude from the roof and absorb solar energy to increase indoor temperature [41]. To obtain more solar radiation, in addition to the use of a glass chimney, one also needs to have a certain angle of tilt. Solar-powered chimneys alone can reduce electricity consumption by 10 to 20% per day [42]. Jing et al. [43] conducted a software simulation study on gaps between 0.2 and 0.6 when a solar chimney was used, and their research results showed that when the gap was 0.5, the airflow rate of flue gas had the best effect. In addition, this also showed that the improved scheme is more consistent with test results.

In addition to the passive heating technology mentioned above, Kang is one of the traditional heating methods used in northern China. Its working principle is using the heat generated by burning biomass to heat a bed, which can effectively improve the local thermal comfort of human body and reduce the demand for energy [44,45]. Similar heating methods have been used in the design of residential buildings in other developed countries, such as Ondol in South Korea and Hypocaus in ancient Rome [46,47]. Traditional Kang is one of the floor heating systems in northern China, and it first appeared in the Zhou Dynasty [48]. It is a rectangular platform composed of bricks, concrete, stone, and other materials. The cavity inside the platform is used as a flue, and the hot smoke generated by cooking utensils is used to add to heated Kang body. Traditional Kang consists of three parts: a stove, Kang body (similar to a bed), and a chimney [49]. A stove is a place where wood is burned [50]. The heat generated during the burning of biomass is used on the one hand for cooking and heating food, and on the other hand, heat is added to Kang body by conveying heat [51]. Traditional Kang heating assembly is composed of two parts: the Kang body and an internal flue. High-temperature flue gas enters the Kang body, flows through the flue, and finally, the flue gas leaves through the chimney. The setting of the flue resistance has a great impact on the flow of flue gas. The greater the flow resistance, the greater the heat conduction time of Kang will be [52]. The surface of the Kang body absorbs the heat and raises the local temperature of the human body through heat transfer processes such as convection and heat conduction [53]. In the rural area of north China, 85% of families use Kang for heating in winter [54]. The research on Kang mainly covers three parts: the development of Kang, the evaluation of the thermal performance of Kang, and the joint application of Kang and other technologies. Yu et al. [55] discussed the development and classification of Kang in terms of cultural background and ethnic divisions; they analyzed the composition, structure, flue, and thermal performance of Kang; and they proposed a new optimized design. The research shows that Kang improves the local thermal comfort of human body; this creates the possibility of influencing indoor temperature and reducing indoor pollutants by optimizing heated Kang equipment. Wang et al. [56] studied and set up a heating wall, tested its thermal performance, and found that the heating wall could store 70% of the heat in a furnace. Through practical measurements and mathematical modeling, it was found that the two results were consistent. This study enhances our understanding of heating walls and provides a reference for rural residential heating design. Chen et al. [57] used a method involving combining solar heating technology and traditional Kang to improve indoor temperature through the investigation of three family houses in the northeast of China. Research results show that when using this technology, the temperature is evenly distributed in the vertical direction, but the initial investment needs to increase by about 10%.

To sum up, the previous research conclusions regarding passive building heating are as follows: Avoid outdoor cold air entering the room (cold enclosure) [58], encourage hot air to enter the room (cold conditioning) [59], and remove indoor local cold air (heat collection) [60]. Cold enclosures are mainly used to change the phase change material of the building envelope structure, affecting the thermal performance of the walls, roof, glass, and other building materials, so as to improve indoor temperature, but the cost is high. Cold conditioning is based on the greenhouse principle and adopts a solar passive design. The indoor temperature is completely determined by the amount of solar radiation entering the building. This method is more suitable for areas with sufficient solar radiation. A combination of solar energy and earth source heat pumps can be used, and equipment units are needed to realize this. Heat collection refers to the use of a fire Kang, but only for heating local parts of human body, rather than for increasing the overall indoor temperature. However, passive heating technology for traditional dwellings should not only consider the construction cost and effect but also consider the production and lifestyle of local people.

### 1.3. Scientific Originalities

Under the influence of the subtropical continental monsoon climate in southern Shaanxi, traditional houses feel wet and cold indoors in winter. Most traditional dwellings are ancient and limited by their construction technology and materials, as there was little consideration for heating measures at the beginning of the design process. To improve indoor temperatures, people use heating equipment such as charcoal, fire ponds, and braziers. However, these types of heating equipment can only improve the indoor local temperature and cannot improve indoor temperature. A passive fire wall heating system is proposed that uses the heat storage capacity of a wall to store the heat generated by smoke when cooking and then release stored heat into the room so as to enhance the indoor temperature.

This kind of heating equipment has three characteristics: cold conditioning, cold enclosure, and heat collection. Cold conditioning is when the heat from cooking heats aluminum tubes inside the walls. Cold enclosure refers to the heat generated by the aluminum tube, which heats the wall, and the wall stores the heat. Heat collection refers to the heat generated by heating the wall, which increases the temperature of the room through thermal radiation and convection. The heating effect completely depends on the heat generated by cooking and does not rely on any mechanical equipment. The increase in the indoor temperature relies on heat radiation and heat convection. Therefore, the heating system does not need any power and is a completely passive heating system. This system is suitable for vast rural areas where the winters are relatively humid and cold. In previous passive heating design research, fewer factors were considered, with the influence of heating technology on indoor temperature being the main aspect analyzed. In this paper, in addition to the heating technology, the influence of solar radiation on indoor temperature is also considered.

### 1.4. Aim of the Study

Through field research, it was found that traditional heating equipment such as hot coals, fire pits, and braziers can increase the local temperature in the room but cannot increase the overall indoor temperature to satisfy the needs of human body. In this study, a fire wall heating system is proposed to improve the indoor temperature, and the effectiveness of the fire wall heating equipment is confirmed using ANSYS software simulation. Its specific objectives are as follows:To investigate, with or without solar radiation on an external wall, the time required for the natural heating system to work as well as the balanced temperature and the heating efficiency.To investigate, with or without solar radiation on an external wall, the time required for the fire wall heating system to work as well as the balanced temperature and the heating efficiency.To evaluate the appropriate time when the open fire wall heating system can work throughout the year.To use the fire wall heating system to increase indoor temperature in an effective timeframe.To investigate the annual heat load reduction outside a wall with or without solar radiation in two cases.To optimize the passive design of traditional dwellings to provide a reference for the design of rural dwellings in the future.

## 2. Methodology

### 2.1. Investigation Research

#### 2.1.1. Location and Climate

The study is located in Zhongshan Village, Zhaowan Town, Xunyang County, Ankang City, Shaanxi Province, which is an area with a hot summer and a cold winter (Figure 1a). Covering an area of about 0.02534 km^2^, it hugs the border crossing road in a belt-like shape, following the natural topography. The village was included in the second batch of villages in the list of traditional Chinese villages. Most of the residential buildings in the village date back to the Qing Dynasty, so the residential buildings are generally one-story or two-story, with the one-story buildings accounting for 85.3% of the total. The floor height of the one-story residential buildings is 4.0–4.8 m, the floor height of the two-story residential buildings is 5.4–6.0 m, and building density is 24.06%. The local climate is humid and mild, with four distinct seasons: a long frost-free period, cold winters with little rain, and summers with drought and rain. Affected by mountainous areas, villages are arranged in clusters, the dwellings are arranged relatively close together, and the streets are generally narrow. Therefore, residential houses are less affected by solar radiation in winter, which makes the air temperature of residential houses lower in winter. The lowest temperature in the village in winter is −3 °C, the highest temperature in summer is 34 °C, and the average wind speed of the whole year is 7 m/s (Figure 1b–d).

#### 2.1.2. Research Method

The research methods of this article are as follows:

Field research: Field research is one of the most basic research methods. In southern Shaanxi, cooking methods, heating methods, thermal performance of residential building materials, and the thermal environment of residential buildings have been deeply understood by means of questionnaire, measurement, and interview. By using the methods of analysis, comparison, and induction, representative traditional folk dwellings were selected as the key research objects.

Data collection: Data collection is mainly the classification and sorting of the current survey data. The field investigation of indoor, outdoor, on the surface of the wall inside, and outside temperature, different cooking indoor temperatures, different heating means of indoor temperature, the temperature of the hearth in the combustion, wall temperature of the flue, doors, windows and the entrance to the position and size, the size of the room size, and other primary data to be screened, classified, and collated.

Induction and summary: Induction and summary is to summarize the data according to certain standards and summarize the properties of the research objects through analysis and comparison. Indoor, outdoor, on the surface of the wall inside and outside temperature, different cooking indoor temperatures, different heating means of indoor temperature, the influence of the human body, the temperature of the hearth in the combustion, and the wall temperature of the flue, doors, windows, and entrances are summarized in relation to the size of the room. The characteristics of thermal environment, cooking mode, heating mode and basic scale of traditional residential buildings in southern Shaanxi are summarized.

#### 2.1.3. Case Study

The research object was traditional residences in southern Shaanxi located in Zhongshan Village (Figure 2). Most of the buildings were one-story, some were two-story, the height of the first story was 3 m, and the height of the second story was 1.8 m (Figure 3a). The walls of the residential buildings were enclosed by blue bricks and the roofs were sloped roofs, which help to block the sun and organize drainage. There was little consideration for village planning in southern Shaanxi. Due to the influence of the mountainous terrain, the arrangement of the dwellings was relatively close, and the roofs had deep overhanging eaves. As a result, residential houses were less affected by solar radiation, so the houses generally felt damp and cold indoors in winter. To improve the temperature of the indoor space, local residents have adopted different heating methods, such as fire ponds, charcoal fire basins, and fire ponds. No matter which method is adopted, its purpose is local heating to increase the surface temperature of human body and the local air temperature, and it is generally used in halls. Traditional heating methods are not used in bedrooms, to prevent carbon monoxide poisoning from occurring during deep sleep at night. By using the heat storage capacity of walls, this method stores the heat of the flue gas generated when cooking and releases the heat energy into the room so as to improve the indoor thermal comfort. Walls with this characteristic are called the walls.

We used three research methods in this case; for the fire wall heating research, we mainly wanted to investigate how to use fire walls to passively improve indoor temperature. Our main idea was to use high-temperature flue gas to heat the pipes of the fire wall during cooking and thus influence indoor temperature through the heat radiation of the fire wall (Figure 3b). In addition, to prevent the high-temperature fire wall from affecting the indoor temperature in summer, a valve should be installed at the stove. Therefore, in the summer, opening the valve will cause smoke to be quickly discharged indoors; in the winter, the valve is closed to heat the flue gas around the wall (Figure 3c,d). When using fire wall heating technology, to improve the heating effect, the pipes must be aluminum and the wall must generally be S-shaped. Using the heat storage and release of the fire wall to improve indoor temperature, the effect of the outdoor temperature on indoor temperature can be decreased.

#### 2.1.4. Analysis of Cooking and Heating Methods

In rural areas of southern Shaanxi, the cooking equipment used in traditional dwellings is generally a wood stove; in contrast, the cooking equipment used in modern dwellings generally includes a gas stove, electric rice cooker, and induction cooker. In our field investigation, it was found that traditional cooking equipment and modern cooking equipment were used together in the same families (Figure 4). The frequency of using wood stoves was 79.3%; the frequency of using gas stoves was 4.1%; and the frequencies of using electric rice cookers and induction cookers were 11.3% and 5.3% (Figure 5a), respectively. It can be seen that the use of wood stoves is still common. The stove is generally located in a corner of the kitchen, and a flue is provided on one side of the stove for smoke exhaust.

In terms of the duration of a single cooking session, the average durations in summer and winter are 60.3 min and 75.2 min, respectively, with small differences between the winter and summer. The field survey results show sessions of 0–30 min for 20.2% and 21.3% of homes in summer and winter, respectively, and sessions of 30–60 min for 34.6% of homes in summer and 36.3% of homes in winter. The time of 60–90 min was taken in the highest proportion of homes, with 45.3% and 36.6% in summer and winter, respectively. Sessions of more than 90 min, mainly in winter, accounted for 5.9% (Figure 5b). It can be seen that the duration of a single cooking session in winter is generally 60–90 min. The main reason for this is that in addition to meeting basic human needs, families must also prepare livestock food in winter, as well as heating their houses and water using a wood stove to reduce the cost of living. Cooking is also related to eating activities. Through field research, we can see that in summer, the proportion of families cooking three meals a day is 86.4%, the proportion of families cooking two meals a day is 10.2%, and the proportion of families cooking one meal a day is 3.4%. In winter, the proportion of families cooking three meals a day is 97.3%, and the proportion of families cooking two meals a day is 2.7% (Figure 5c). From this, we can deduce that most families in this area cook three meals a day. Field research was conducted on the times every day when the three meals are cooked. In winter, 89.5% were cooked in the period 8:30–9:30 in the morning, 91.6% were cooked in the period 11:30–12:30 in the afternoon, and 88.6% were cooked in the period 5:30–6:30 in the evening (Figure 5d).

Traditional heating methods include the use of a fire pit, charcoal fire basin, fire pond, etc., which can also be used for cooking; the main use of firewood and charcoal is as an open heat source, causing indoor pollution during combustion, which is a serious issue. Modern heating equipment uses electric energy—for example, electric heating fans, foot warmers, and electric ovens. Additionally, baking tables burn biomass. In the field survey, 69.3% of residences used firewood for heating energy, 16.7% used carbon, 6.7% used coal, and 7.3% used electricity. The test time was December 26, 2020, and the test location was the traditional residences of Zhongshan Village in southern Shaanxi. The test objects were 5 items: fire ponds, charcoal braziers, fire tables, wood stoves, and electric fans. The instrument used for testing was an infrared imager (Type HM-TPH11-3AXF), which was used to measure the temperature distribution of cooking equipment and the temperature of the surface of human body. The range of the instrument parameters was −20 to 350 °C, and the sensitivity was 0.5 °C. The measuring position was 1.5 m away from the cooking equipment and human body. When measuring the data, we tried to keep the object and human body in a stable state and take pictures and measurements at a distance of 1.5 m from the ground. By using an infrared thermal imager to test the temperature distribution of traditional and modern heating equipment, we found that the average heat source temperature of traditional heating equipment is generally higher than that of electric heating equipment. The average fire pond temperature was 103.68 °C, and the highest temperature was 245.68 °C (Figure 6a). The average temperature of the charcoal brazier heat source was 203.43 °C, and the highest temperature was 241.24 °C (Figure 6c). The average temperature of the furnace body of the baking table was 111.32 °C and the highest temperature was 142.73 °C; the average temperature of the table top was about 25.31 °C (Figure 6e). The firewood stove was also one of the indoor heat sources analyzed. The average temperature of the stove wall was 110.34 °C, the highest temperature was 232.32 °C, and the average temperature of the pot body was 156.35 °C (Figure 6g). The highest temperature of the heating element of electric heating equipment was 231.36 °C, but the average temperature was only 50.26 °C (Figure 6i). The surface area of the heat source was generally smaller than that of the traditional heating equipment.

Using an infrared thermal imager to test the temperature distribution on the human body surface when near heating cooking equipment, it was found that the average surface temperature of the human body when facing away from the heat source was 37.02 °C/10.52 °C (fire pond) (Figure 6b), 32.93 °C/14.32 °C (charcoal brazier) (Figure 6d), 18.03 °C/7.07 °C (fire table) (Figure 6f), 6.02 °C/4.21 °C (wood stove) (Figure 6h), and 36.68 °C/13.96 °C (electric fan) (Figure 6j). Comparing non-open heat sources (roasting tables, wood stoves) with using open heat sources (fire ponds, charcoal braziers), the body surface temperature rises more. The average temperature differences between the side facing the heat source and the side facing away from the heat source differ by values of 26.5 °C (fire pond), 22.72 °C (electric fan), 18.76 °C (charcoal basin), 10.96 °C (fire table), and 1.81 °C (firewood stove). In a word, the heating equipment used in rural areas of southern Shaanxi focuses on local heating as its goal. No matter what type of heating equipment is used, it will focus on local heating, improving the local surface temperature of human body (especially in the limbs) and the indoor local temperature.

### 2.2. Passive Heating Design

#### 2.2.1. Design of Cooking Heating System

The building is oriented north–south, and the plane function is composed of three parts: bedroom, living room, and kitchen. The building has three rooms with two depths, and the wall thickness is 240 mm. The east side is composed of a kitchen and a bedroom, the middle is used as the living room, and the west side has two bedrooms. The transformed cooking heating system scheme is as follows: Between the bedroom and kitchen on the east side, a heating wall (i.e., fire wall) is installed to raise the indoor temperature. The red area in the picture indicates the location of the fire wall (Figure 3a). The specific structure is that a smoke exhaust port with a diameter of 100 mm is set inside the stove, and a 2 mm thick aluminum pipe is placed in the smoke exhaust port. To increase the heating area, the aluminum tube is placed in an “S” shape with a total length of 11.68 m. Its major function is to use the high temperature of the fire wall to increase the indoor temperature. First of all, the high-temperature flue gas heats the aluminum tube through thermal convection, and the heated aluminum tube heats the fire wall through heat conduction and thermal radiation. Between the walls, convection heat exchange and thermal radiation are used to increase the temperature of the wall, thus increasing indoor temperature. Secondly, we compare the wall with or without solar radiation for the cooking heating system. This new cooking heating system can raise the overall temperature compared to the previous heating methods, rather than just offering local heating.

#### 2.2.2. Working Principle of Fire Wall

The working principle of the fire wall is using the heat storage capacity of the wall to store the heat generated by flue gas during cooking and releasing the accumulated heat indoors, thereby improving the indoor thermal comfort to a certain extent. The fire wall can be regarded as a cooking equipment heat recovery device. The whole heat transfer process is affected by many factors, such as the energy density of the combustion, the material properties of the wall, the material and size of the pipe, the size of the room, the indoor temperature and humidity, sunshine, and other factors. To facilitate understanding, this heat transfer process is divided into three parts: First of all, the burning material of the stove produces high-temperature flue gas, and flue gas flows through the aluminum tube, thereby forming turbulence. In the software simulation, the turbulence intensity needs to be calculated. The formula is as follows:(1)$$I\;=\;0.16{\mathrm{re}}^{\left(-1/8\right)}$$
where I is the turbulence intensity and re is the Reynolds number.

This flow process heats the aluminum tube by a heat transfer process dominated by heat convection, and the heat transfer process is divided into three parts. The turbulence generated by the flue gas in the aluminum tube causes transient heat transfer. After a period of time, the steady-state heat transfer is the mainstay. At the end of the cooking, the transient heat transfer is the mainstay. This heat transfer is a fluid–solid coupling transfer of heat. In the process of convective heat transfer, we follow the basic calculation formula—Newton’s law of cooling.
(2)Φ = qA = AhΔt = Δt/(1/hA)
where Φ is the heat transfer power, q is the heat flux density, convection heat transfer coefficient h is material, A is the heat transfer area, 1/hA is the convective heat resistance, and ∆t is the surface temperature and environmental temperature difference value.

A stable heat pipe heats the wall through heat conduction and heat radiation. Its heat transfer belongs to solid–solid coupling heat transfer, which follows Fourier’s Law in the process of heat conduction. The calculation formula is as follows:(3)Q/A = q 
where Q is the heat transfer rate, q is the heat flux density, and A is the heat transfer area.

Secondly, the heated wall heats space between the walls through convection heat transfer and heat radiation, thus improving indoor temperature. This heat transfer process is coupled with fluid and solid heat transfer. In the process of thermal radiation, Stefan–Boltzmann’s law is followed, and the calculation formula is as follows:(4)$$M(T)\;=\;\sigma T^{4}$$
where M(T) is the radiation energy, T is the absolute temperature of the object, and σ is the constant 5.67 × 10^−8^.

When cooking is stopped, the temperature of the aluminum tube begins to drop, thus stopping the heating of the fire wall.

#### 2.2.3. Whether the Wall Has Natural Heating with Solar Radiation

In the winter, a natural heating system relies mainly on solar radiation to raise the temperature of the interior through the walls. The heat transfer process mainly includes three parts: the first is that the wall surface absorbs heat, and the outer surface of the wall absorbs heat energy through convection heat exchange and thermal radiation. The second is the heat transfer of the wall itself, which transfers from the high-temperature outer surface to the low-temperature inner surface. Finally, the inner surface radiates heat, and the inner surface of the wall radiates heat to an indoor environment. The heat transfer method of each part is a comprehensive process of heat transfer, convection, and heat radiation. In the case of natural heating, the difference between the indoor temperature and the outdoor temperature is 2.5 °C (Figure 8i).

Without solar radiation indoors, the heat source mainly depends on the indoor heat disturbance source. The indoor heat disturbance source mainly refers to the heat generated by electrical appliances and human body. In addition to the influence of indoor thermal disturbance, the influence of indoor temperature changes is also related to the building scale, window size, and ventilation frequency. However, in past passive heating design research, the lack of solar radiation was not considered. To a certain extent, this makes software simulations more accurate and usable compared with reality. In the case of natural heating without considering solar thermal radiation, the indoor temperature changes with the outdoor temperature and the change range is small, with an indoor temperature to outdoor temperature difference of 0.2 °C.

#### 2.2.4. Whether the Heated Wall Has Solar Radiation

To increase indoor temperature, the heat generated during cooking is used to heat the wall, and the heat absorption and heat storage capacity of wall is used to increase indoor temperature and reduce the effect of the outside temperature. The fire wall is a double-sided heating wall that can simultaneously use heat radiation to heat the rooms on both sides of the fire wall. The effect of flue gas entering the pipe wall and heating the pipe wall is shown in Figure 9c–f. When the heat pipe heats the fire wall, the temperature of the fire wall increases significantly (Figure 10a–f). A period of time after the fire wall becomes hot, the temperature of the bedroom rises significantly compared to in natural heating. Without solar heat radiation, the cooking system stops heating when the heating time is 48 min, and the indoor temperature increases (Figure 11k). Using a cooking heating system, in the case of solar heat radiation, heating is stopped when the heating time is 48 min, and the heating effect of residential buildings is better (Figure 11m). The temperature after equilibrium and the time spent were improved, and the heat load of the building was also reduced.

### 2.3. Simulation Process

The software used in this simulation was Ansys, Openstudio, Ladybug, Energyplus. The boundary condition is whether the wall has solar heat radiation or not. The software Ansys mainly calculates the heating time of the room and final temperature of the room after heating. The software products OpenStudio, Ladybug, and EnergyPlus are mainly used to calculate the thermal load reduction. The software Ansys was mainly used for turbulence simulation, steady-state thermal power, transient thermal power, and thermal radiation simulation calculation. Turbulence simulation mainly calculates the time taken for flue gas to pass through the aluminum tube, the wind vector of the flue gas, and the temperature of the tube wall. The steady-state thermal power and transient thermal power are mainly used to calculate the temperature of the heat pipe heating the wall and the time required for this. Thermal radiation simulation is mainly used to calculate the thermal radiation of the fire wall and the heat exchange between walls. Heating load after adopting the new heating system is calculated and simulated using OpenStudio, Ladybug, and EnergyPlus. The steps are as follows: the model was established in OpenStudio, the model was imported into the ladybug to set relevant parameters, and the heat load was calculated using EnergyPlus.

The simulation was divided into three parts. The first part was the simulation of natural heating. The model was established by measuring data, and the local winter climate conditions data were used, including temperature and humidity, wind speed, sunshine, and so on. The simulation content includes the temperature of the room equilibrium and the time required. The second part is to simulate the heating of the fire wall. Firstly, we establish the fire wall system, which mainly includes the hearth, flue, and fire wall. The simulation content is the velocity and temperature of flue gas, the heating time of the fire wall, and the indoor temperature and the required time after equilibrium. The third part is the simulation calculation of the appropriate time of the fire wall system and the reduction in heat load. The basic data statistics are as follows: Table 1 shows the basic information of residential buildings and the construction methods and materials used for walls, windows, floors, roofs, and other structures. Table 2 shows the heat storage coefficient, specific heat capacity, and other thermal indexes of different structural materials.

## 3. Results and Discussion

### 3.1. Subsection Comparison of Natural Heating and Wall with or without Solar Radiation

#### 3.1.1. Heating Time and Final Equilibrium Temperature

To better verify the effectiveness of the cooking heating system, comparisons were made with the natural heating effect as a benchmark. The construction materials of the folk houses in southern Shaanxi include cement mortar, clay brick masonry, lime mortar, gravel, gravel, pebble concrete, blue brick, waterproof membrane, wood, and aluminum. The specific simulation method is as follows: use the Ansys software to simulate, import the model into Ansys, set the grid, activate the energy equation, select the turbulence model, and set the radiation model. Add the thermal properties of the above materials, set the boundary conditions, and solve the method. Finally, the flow field is initialized to solve the simulation process. Natural heating mainly relies on solar radiation through the wall, glass, roof, and other structures to improve the indoor temperature. Through comparison with multiple sets of measurement data, the day when the outdoor temperature is least affected by solar radiation in winter is selected, and the temperature distribution is shown in Figure 7. First of all, from 18:00 to 9:00 is the time period that is least affected by solar radiation 24 h a day, and the difference between the indoor temperature and outdoor temperature is small. However, with the increase in solar radiation from 9:00 to 18:00, the wall began to be affected by solar radiation, and the indoor temperature began to change. Therefore, the simulation time was selected from 9:00 to 18:00. It can be seen from the simulation calculation that from 9:30 to 11:30, as the sun’s altitude angle changes, the temperature of the room on the first floor on the east side and the meeting room in the middle begins to increase from the initial average temperature of 6 to 7 °C (Figure 8a,c). The temperature of the rooms on the second floor on the east side rises slowly, and the four rooms on the west side are less affected by solar radiation, so the temperature is slightly reduced. From 11:30 to 14:30, the range of the indoor temperature change is small, the indoor temperature tends to be stable, and the average indoor temperature is 7 °C (Figure 8e). From 14:30 to 18:00, the solar radiation is mainly distributed in the west side of the room (Figure 8g). At 16:30, the room on the first floor in the west side reaches a maximum temperature of 7.5 °C, and this then begins to decline (Figure 8i). At 18:00, the temperature of the room on the east side decreases due to the influence of solar radiation, and the temperature of the room begins to drop, from 7 to 6.5 °C (Figure 8k). In the case of natural heating with solar radiation, the indoor mean temperature is 7 °C, which is a 2.5 °C difference from the outdoor average temperature.

Through software simulation calculation, the indoor temperature is basically the same as the outdoor temperature in the case of natural heating without solar radiation. The simulated time is the same as the natural heating with solar radiation, and the time is also from 9:00 to 18:00. From 9:00 to 12:00, the average temperature of the first and second floors indoors is 4.5 °C, mainly because the outdoor temperature is relatively low (Figure 8b,d,f). From 12:00 to 14:00, the indoor temperature is basically the same. From 14:00 to 17:00, the outdoor temperature begins to rise slowly, and the indoor temperature of the first floor also begins to rise slowly (Figure 8h,i). After equilibrium, the temperature is 4.8 °C, and the second floor remains unchanged. The temperature begins to drop from 17:00 to 18:00, and the temperature on the first and second floors is 4.5 °C (Figure 8m). The research results show that in the case of natural heating without solar radiation, the mean indoor temperature is 4.7 °C and the temperature change range is small, with only a 0.2 °C difference from the average outdoor temperature.

#### 3.1.2. Comparative Analysis

In natural heating with or without solar radiation, the difference between the indoor temperature and the outdoor temperature is small. In the case of solar radiation, the heating time is 4.30 h; the equilibrium temperature is 7 °C, which is a 2.5 °C difference from the outdoor value; and the heating efficiency is 0.22 °C/h. Without solar radiation, the heating time is 6.0 h. The temperature after equilibrium is 4.8 °C, which is a 0.2 °C difference from the outdoor temperature. This is basically in equilibrium with the outdoor temperature, and the heating efficiency is 0.03 °C/h. The heating effect of solely relying on solar radiation is poor, and the heating time is longer. In the process of natural heating, the heat generated by solar radiation first heats the wall, and the heat passes through the wall to heat the room. This heating method is greatly affected by the weather, the heating efficiency is extremely low, and the effect of improving the indoor temperature is not good.

### 3.2. Comparison of Cooking and Heating with or without Solar Radiation on the Wall

#### 3.2.1. Heating Time and Final Equilibrium Temperature

In comparing the simulation results with the results of natural heating, we can see that the outdoor temperature distribution is the same as the natural heating outdoor distribution and the initial indoor temperature is also 6 °C (Figure 7). The process of cooking and burning is divided into four stages: material ignition, open flame ignition (continuous fuel supply), smokeless combustion (closed flue), and material ignition. After the measurement, the flue gas enters the aluminum tube at an average speed of 2.5 m/s after the material is ignited. Through Ansys simulation, it can be seen that the speed at the turn is 1.6 m/s and the wind pressure is 1.1 pa (Figure 9a). The speed of the flue gas to the smooth part of the aluminum tube is increased to 3.2 m/s, and the wind pressure is 1.4 pa (Figure 9b). Finally, it is discharged from the aluminum tube at a speed of 2.6 m/s. The change in the flue gas velocity is mainly caused by the hot pressure generated by the flue gas in the aluminum tube and the shape of the aluminum tube. The temperature of the flue gas depends on the amount of heat generated when the material is burned. After many measurements, the temperature of the flue gas is generally 60~70 °C. The flue gas enters the aluminum tube at a temperature of 60 °C. At 38 s, the inlet of the aluminum tube is 12.65 °C, and the temperature of the remaining aluminum tubes is maintained at 2 °C (Figure 9c). At 84 s, the average temperature of the aluminum tube is 36.94 °C (Figure 9d). At 102 s, the average temperature of the aluminum tube is 42.55 °C (Figure 9e). At 134 s, the aluminum tube reaches a constant temperature of 60 °C (Figure 9f). Since the thermal conductivity of the aluminum tube is higher than 203 W/m· k, the temperature of the aluminum tube increases faster.

After the aluminum tube is heated, the heat wall is heated by heat conduction and thermal radiation. At 5 min and 15 s, the average temperature of the fire wall is 9.07 °C, which is mainly distributed around the aluminum tube, and the lowest temperature on the ground is 5.94 °C (Figure 10a). At the time of 8 min and 30 s, the average temperature of the fire wall is 12.15 °C, and the lowest temperature at the ground is 5.88 °C (Figure 10b). With the increase in heating time, the average temperature of the fire wall reaches 15.22 °C at 12 min and 45 s, representing an average increase of 2 °C (Figure 10c). At 16 min and 27 s, the temperature of the fire wall gradually increases. The average temperature of the fire wall is 21.44 °C, and the lowest temperature at the ground is 5.67 °C (Figure 10d). At 20 min and 24 s, the average temperature of the fire wall is 24.51 °C (Figure 10e). At 25 min and 35 s, the temperature of the fire wall no longer changes, reaching a constant value of 30.78 °C, and the lowest temperature at the ground is 5.24 °C (Figure 10f).

From the above survey on the length of a single cooking time, it can be known that 60–90 min is the main one. To make the simulation more realistic, we chose a single cooking time of 75 min. When the fire wall temperature reaches a constant value, it takes 27 min and 49 s to cook. There should be 48 min and 11 s remaining for the fire wall to heat up. To simulate the influence of the fire wall on the indoor temperature through convective heat transfer and thermal radiation, the simulation time started at 9 o’clock, and the initial indoor temperature was 6 °C. With solar radiation, at 9:05, the temperature of the first and second floor rooms on both sides of the fire wall began to rise from 6 to 7 °C (Figure 11a). The temperature of the living room and the other rooms remained basically unchanged. At 9:10, the temperature in the rooms on both sides of the wall rose to 7.8 °C, while the temperature in other rooms rose to 6.5 °C (Figure 11c). At 9:20, the room temperature on both sides of the fire wall was 9.5 °C, and the temperature in the other rooms was 7.2 °C (Figure 11e). At 9:30, the temperature of the room on both sides of the fire wall was 11.5 °C, and temperature of the other rooms was 7.9 °C (Figure 11g). At 9:40, the temperature of the room on both sides of the fire wall was 13.6 °C, and the temperature of the other rooms was 8.7 °C (Figure 11i). At 9:48, the temperature of the rooms on both sides of the fire wall was 14.6 °C, the temperature of other rooms was 9.8 °C, and the fire wall heating was stopped (Figure 11m).

With the closure of the fire wall cooking system, the indoor temperature began to slowly drop. Within 30 min of stopping cooking, the indoor temperature was in a stable state. The temperature of the first and second floors on both sides of the fire wall was still maintained at 14.6 °C, and the temperature in the other rooms was maintained at 9.8 °C. The mean indoor temperature was 12.2 °C. The temperature increased by 6.2 °C. Thirty minutes after stopping cooking, the temperature of the rooms on the second floor on both sides of the fire wall dropped to 11.2 °C, the temperature of the room on the first floor was 10.5 °C, and the temperature of the other rooms dropped to 8 °C (Figure 12a). Forty-five minutes after cooking was stopped, the temperature of the second-floor room on both sides of the fire wall dropped to 10.2 °C, the temperature of the first-floor room was basically in equilibrium with the temperature of other rooms, and the temperature was 8 °C (Figure 12c). One hour after stopping cooking, the indoor temperature was steady at 8 °C, but due to the influence of solar radiation, the temperature in the east room was 8.5 °C (Figure 12e). To prove the effectiveness of the fire wall heating system, simulations were carried out without solar radiation. The simulation time also started at 9 o’clock. At 9:05, the temperature of the rooms on both sides of the fire wall began to rise, the temperature was 6.5 °C, and the temperature of other rooms was 6 °C (Figure 11b). At 9:10, the temperature of the rooms on both sides of the fire wall was 7 °C and the temperature of the other rooms remained unchanged (Figure 11d). At 9:20, the temperature in the first and second floor rooms on both sides of the fire wall was 7.5 °C, while the temperature in the other rooms began to drop to 5.5 °C (Figure 11f). This happened because there was no solar radiation and the outdoor temperature dropped, so the temperature of the rooms without a fire wall system began to drop. At 9:30, the temperature of the rooms on both sides of the fire wall was 9.3 °C, and the temperature of the other rooms was 6.5 °C (Figure 11h). This situation occurred because when the temperature of other rooms is in balance with the outdoor temperature, the increase in the temperature of the rooms on both sides of the fire wall affects other rooms, causing the indoor temperature to rise. At 9:40, the temperature of the rooms on both sides of the fire wall was 10.5 °C, and the temperature of the other rooms was 7.1 °C (Figure 11j). At 9:48, the temperature in the rooms on both sides of the fire wall was 11.8 °C, and the temperature in the other rooms was 7.6 °C (Figure 11m). When the heating system of the fire wall was stopped for 20 min, the indoor temperature on both sides of the fire wall began to drop to 8 °C, and the average temperature of the other rooms was 6.0 °C (Figure 12b). Thirty-five minutes after the heating was stopped, the average indoor temperature was 5.5 °C (Figure 12d). Fifty minutes after the heating was stopped, the average indoor temperature was maintained at 5.5 °C (Figure 12f).

#### 3.2.2. Comparative Analysis

Due to the cooking and heating system, the cooking time is limited. The early stage of the fire wall system requires 27 min and 49 s, so the heating time is 48 min and 11 s. In the case of solar radiation, the indoor temperature remains basically unchanged for 30 min after the cooking activity stops. The temperature of the rooms on both sides of the fire wall is 14.6 °C, and the other rooms are kept at 9.8 °C. The mean indoor temperature is 12.2 °C, and temperature increases by 6.2 °C. Without solar radiation, the indoor temperature remained basically unchanged for 20 min after cooking stopped. The temperature of the first and second floors on both sides of the fire wall was 11.8 °C, and that of the other rooms was 7.6 °C. The average indoor temperature was 9.7 °C, which increased by 3.7 °C.

Natural heating systems mainly rely on solar radiation to generate heat in the walls to affect the interior temperature. However, when a cooking heating system is heated, it mainly relies on a fire wall for heating, which greatly increases the heating efficiency. Without considering solar radiation, the average heating efficiency for cooking and heating is 2.34 °C/h. Considering solar radiation, the average heating efficiency for cooking and heating is 3.54 °C/h.

### 3.3. Suitable Time for Applying New System

The heating system, whether with or without solar radiation, has an effect on the heating time, heating efficiency, post-heating equilibrium temperature, and reduced heat load. The time taken to use the cooker for heating, changes in indoor temperature, and reduced heat load were estimated with and without solar radiation throughout the year. The most suitable time to use the heating system is mainly in January, February, March, November, and December. Due to the limitations of cooking heating time, in the case of solar radiation, the annual use of cooking heating system is 667 h, accounting for 7.71% of the total time of the year (Figure 13a). In the case of there being no solar radiation, the annual use of cooking heating system is 424 h, accounting for 4.91% of the total time of the year (Figure 13b).

### 3.4. Total Thermal Load Reduced by the New System

Using Energyplus to simulate the annual heat load of the cooking and heating system, when the wall experiences solar radiation, the annual heat load of the building is reduced by 1104.723 kW· h, with a reduction rate of 19.84% (Figure 14a). When there is no solar heat radiation on the wall, the annual heat load of the building will be reduced by 440.8318 kW· h, which is a reduction rate of 7.91% (Figure 14b). The thermal load difference between the two is 663.8912 kW·h.

### 3.5. Room Temperature below 18°

Figure 15 shows the variation in indoor temperature with or without solar radiation when the building adopts a natural heating system. We take 18 °C as the limit of the passive heating of the building and calculate the time below 18 °C. In the case of natural heating, the total annual time when the temperature in the building is below 18 °C is 4430 h, accounting for 51.27% of the total annual time (Figure 15a). When the cooking system is used and the wall experiences solar radiation, the annual time when the temperature in the building is lower than 18 °C is 3763 h, accounting for 43.55% of the total time of the year (Figure 15b). Without solar radiation on the wall, the total annual time when the building is below 18 °C is 4006 h, accounting for 46.36% of the total annual time (Figure 15c).

## 4. Conclusions and Future Research

### 4.1. Conclusions

This work studied the passive heating of traditional houses in southern Shaanxi in winter and proposes a cooking fire wall heating system. First, a model of the cooking fire wall was designed to use the waste heat from cooking to increase the indoor temperature. Secondly, this paper studied the indoor temperature change with or without solar radiation when using the cooking fire wall system and natural heating. Finally, the results of using the cooking fire wall system and natural heating to increase the indoor temperature are compared with or without solar radiation. The indoor temperature changes of the kitchen fire wall system were evaluated under the conditions of whether there was solar radiation throughout the year or not. Without solar radiation, the cooking fire wall system can increase the indoor temperature. The specific conclusions are as follows:In order to increase the indoor temperature of traditional dwellings in southern Shaanxi in winter in accordance with local living customs, a cooking fire wall system was designed. Using the waste heat generated by cooking and the characteristics of wall heat storage, the heat was transferred to the room through the system. A software simulation confirmed that the new system caused a significant increase in the indoor temperature, and it specifically increased by 3.7 °C.By simulating natural heating, without solar radiation, the indoor temperature will eventually reach 4.8 °C, and the heating efficiency will be 0.03 °C/h. With solar radiation, the indoor temperature will eventually be 7 °C, and the heating efficiency will be 0.22 °C/h.Through the simulation of the cooking fire wall system, without solar radiation, the indoor temperature is increased to 9.7 °C, and the heating efficiency is 2.34 °C/h. With solar radiation, the indoor temperature is significantly increased to 12.2 °C, and the heating efficiency is 3.54 °C. Compared with natural heating, the heating effect of the cooking fire wall is obvious.Using the cooking and heating system, the heat load of the building was reduced by 1104.723 kW·h (reduction rate 19.84%) and 440.8318 kW·h (reduction rate 7.91%) throughout the year with or without solar radiation. In addition, with 18 °C as the limit of the passive heating of buildings, the total heat load time of the building throughout the year with or without solar radiation accounted for 43.55% and 46.36% of the total time of the year, respectively.

### 4.2. Future Research

Future research should mainly focus on the following aspects:The results of this study need to be proved in practice. Future research should focus on creating a detailed method of construction for the fire wall system. Additionally, the thermal properties of the fire wall material need to be considered, such as the insulation and air tightness of the material. It is necessary to maximize the heat storage performance of the fire wall, thereby extending the heating time.Consideration could be given to building a more comprehensive fire wall system, connecting the various rooms, and discussing indoor temperature changes and the reduction in the overall thermal load of the building.Technology for combining the cooking system with a solar chimney can be considered to analyze the temperature changes and extend the heating time.

## Figures and Tables

**Figure 1 ijerph-18-03745-f001:**
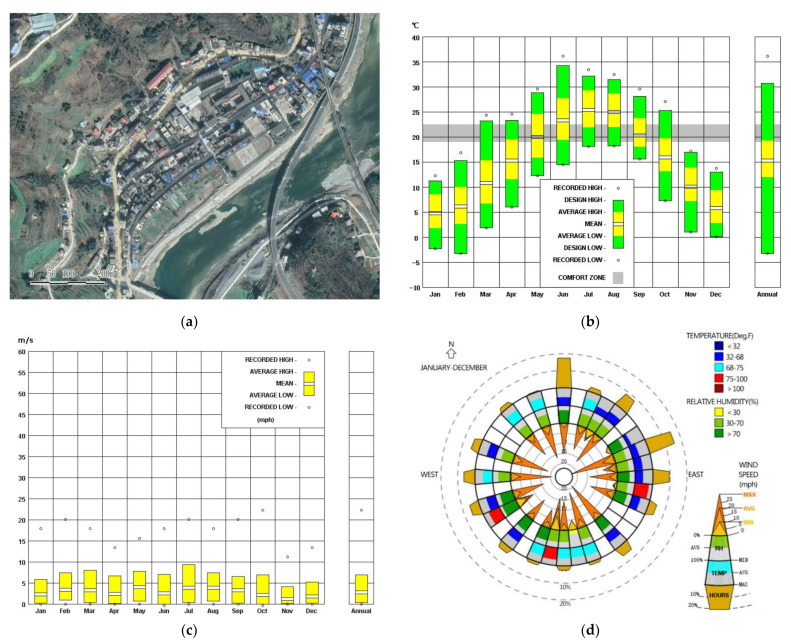
(**a**) Village satellite map; (**b**) Average annual temperature; (**c**) Average annual wind speed; (**d**) Temperature, humidity, and air velocity distribution throughout the year.

**Figure 2 ijerph-18-03745-f002:**
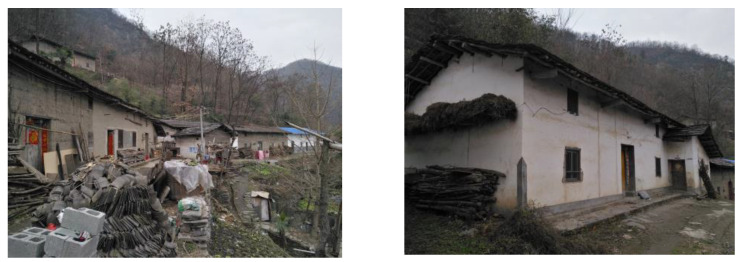
Photo of the present situation of folk houses in southern Shaanxi.

**Figure 3 ijerph-18-03745-f003:**
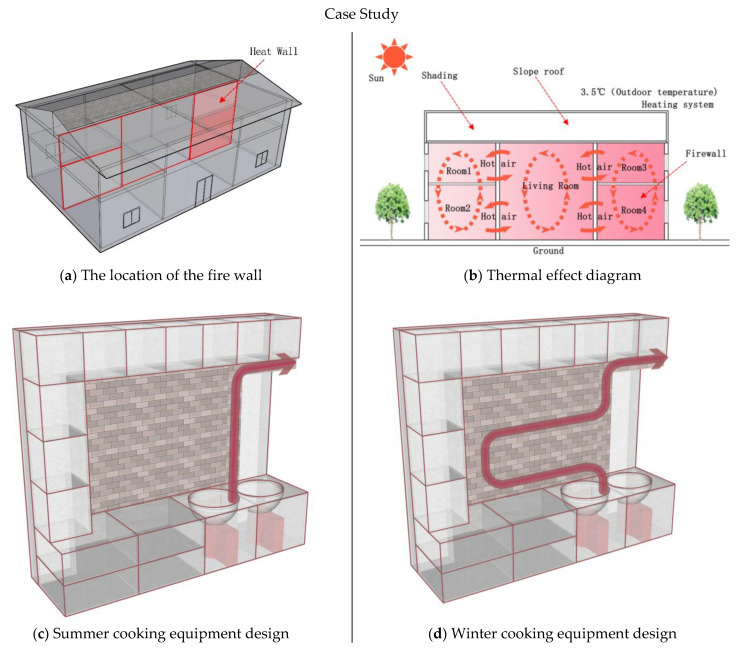
(**a**) The location of the fire wall; (**b**) Thermal effect diagram; (**c**) Summer cooking equipment design; (**d**) Winter cooking equipment design.

**Figure 4 ijerph-18-03745-f004:**
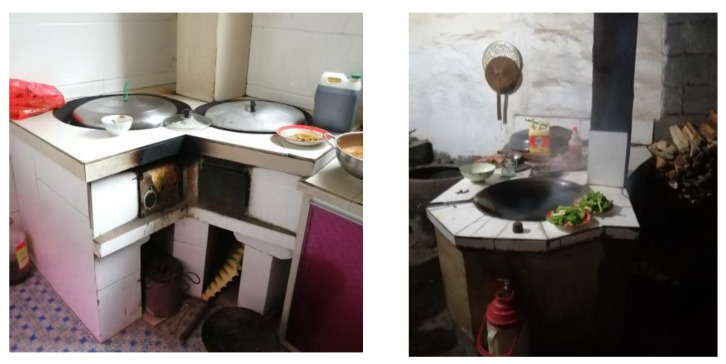
Photo of the present situation of cooking in southern Shaanxi.

**Figure 5 ijerph-18-03745-f005:**
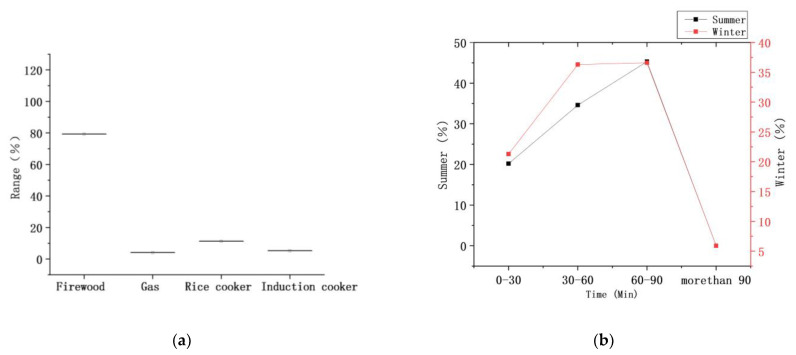
(**a**) Frequency of using cooking equipment; (**b**) Time taken for a single cooking session; (**c**) Number of meals a day; (**d**) Time taken for cooking three meals a day.

**Figure 6 ijerph-18-03745-f006:**
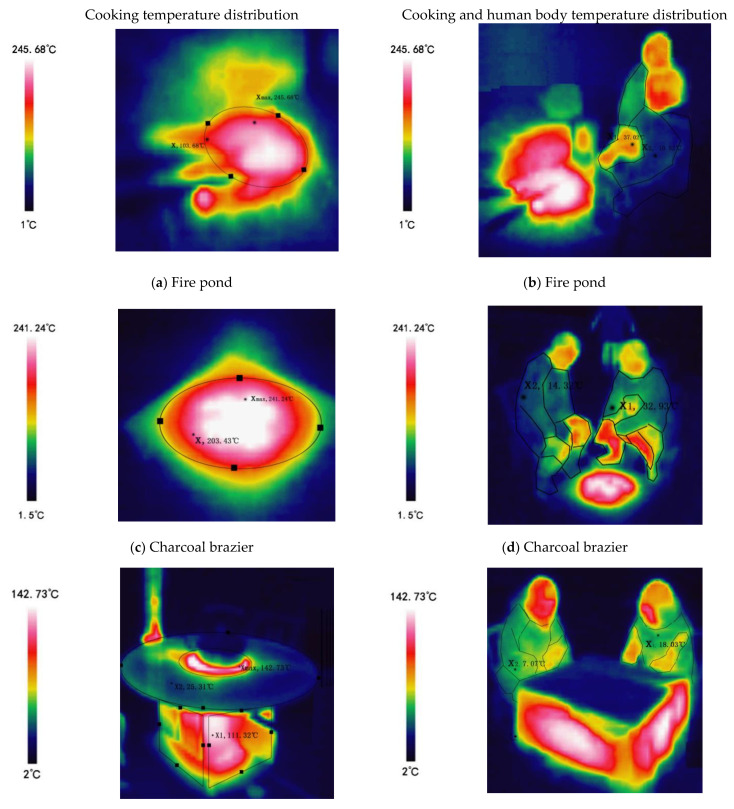
(**a**) Fire pond temperature distribution; (**b**) Fire pond and human body temperature distribution; (**c**) Charcoal brazier temperature distribution; (**d**) Charcoal brazier and human body temperature distribution; (**e**) Baking table temperature distribution; (**f**) Baking table and human body temperature distribution; (**g**) Firewood stove temperature distribution; (**h**) Firewood stove and human body temperature distribution; (**i**) Heating element temperature distribution; (**j**) Heating element and human body temperature distribution.

**Figure 7 ijerph-18-03745-f007:**
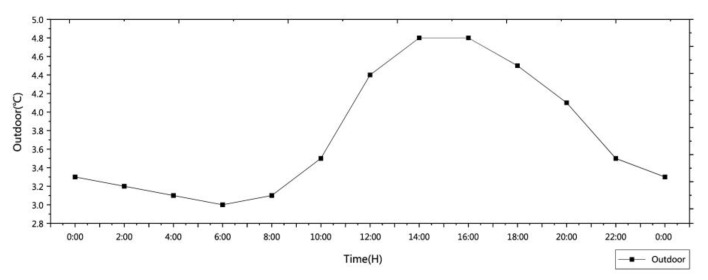
Outdoor temperature distribution.

**Figure 8 ijerph-18-03745-f008:**
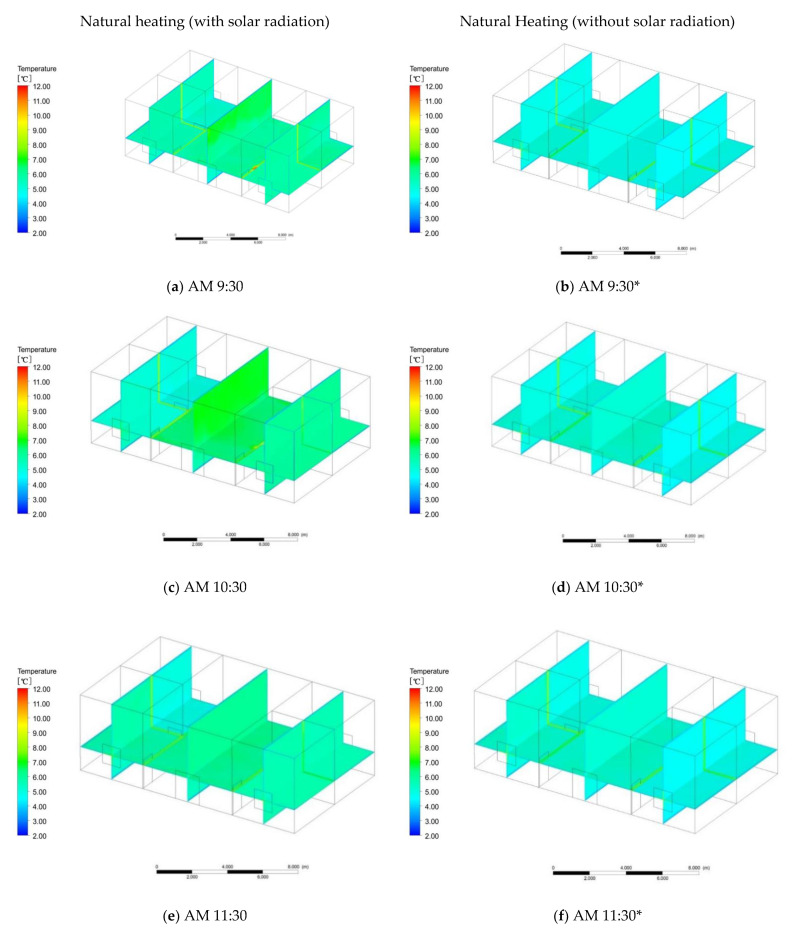
Natural heating at defined times; * refers to heating without solar radiation.

**Figure 9 ijerph-18-03745-f009:**
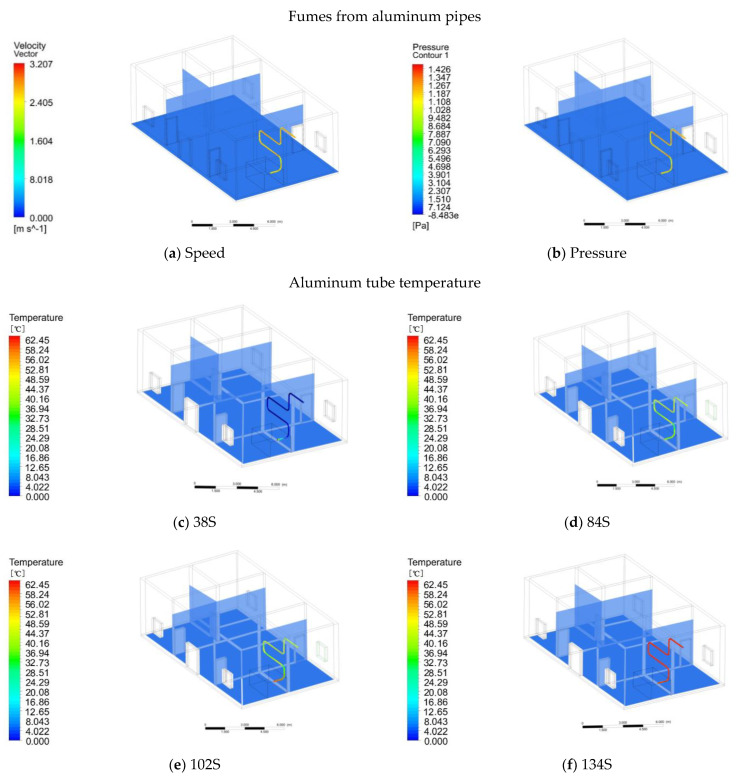
(**a**) Flue gas velocity in aluminum tube; (**b**) Flue gas pressure in aluminum pipe; (**c**) The temperature of 38-s aluminum tube; (**d**) The temperature of 84-s aluminum tube; (**e**) The temperature of 102-s aluminum tube; (**f**) The temperature of 134-s aluminum tube.

**Figure 10 ijerph-18-03745-f010:**
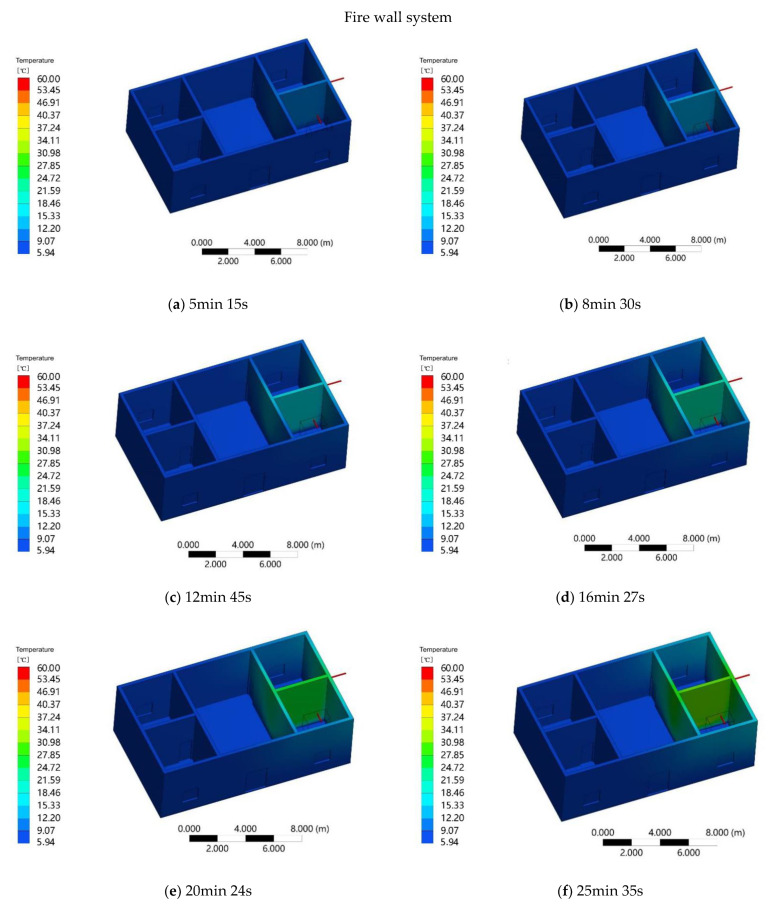
Wall heating for specific times.

**Figure 11 ijerph-18-03745-f011:**
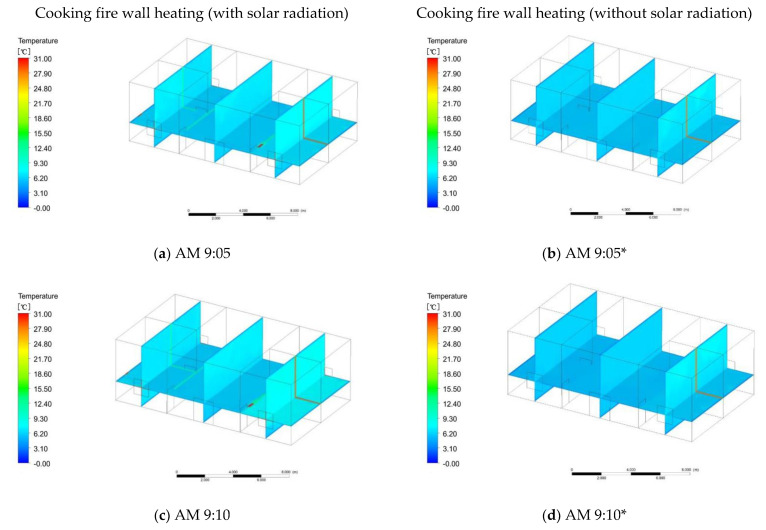
Cooking and heating at defined times; * refers to heating without solar radiation.

**Figure 12 ijerph-18-03745-f012:**
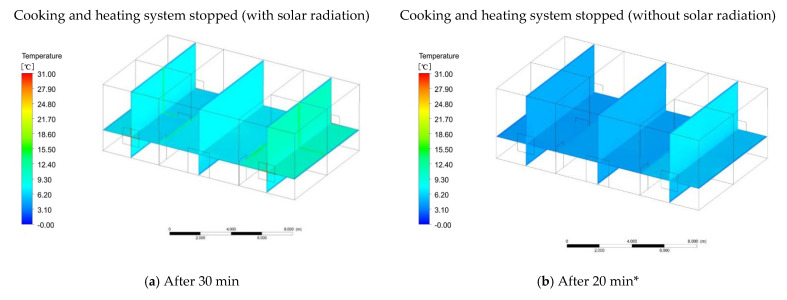
Temperature after the heating system was powered off; * refers to heating without solar radiation.

**Figure 13 ijerph-18-03745-f013:**
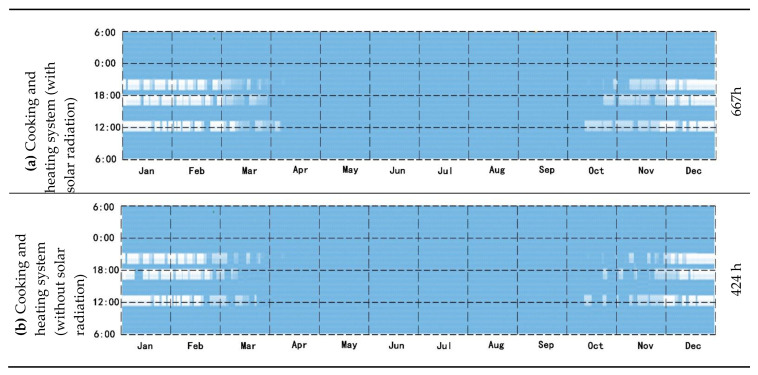
(**a**) Suitable time of the annual average for cooking and heating system with solar radiation; (**b**) Suitable time of the annual average for cooking and heating system without solar radiation.

**Figure 14 ijerph-18-03745-f014:**
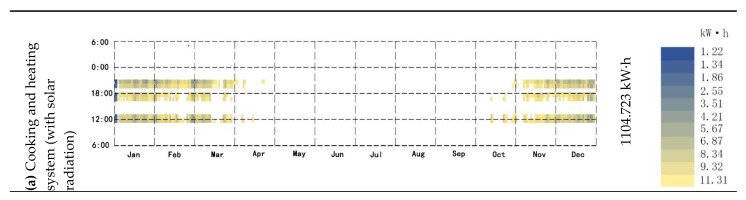
(**a**) The annual average load reduction in the cooking and heating system with solar radiation; (**b**) The annual average load reduction in the cooking and heating system without solar radiation.

**Figure 15 ijerph-18-03745-f015:**
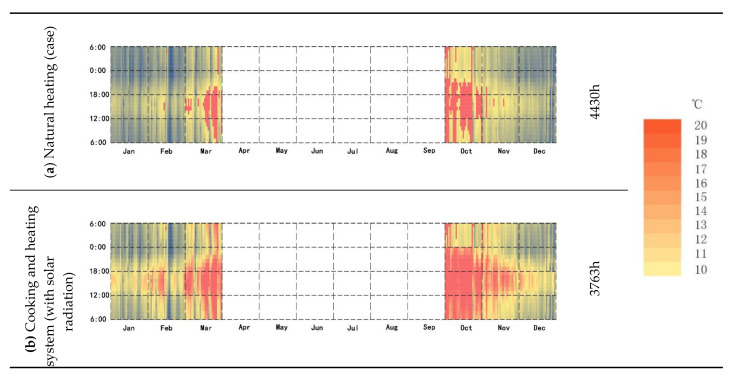
(**a**) The time when the average annual temperature of natural heating is lower than 18 °C; (**b**) Annual time when using cooking heating with solar radiation when the average temperature is below 18 °C; (**c**) Annual time when using cooking heating without solar radiation when the average temperature is below 18 °C.

**Table 1 ijerph-18-03745-t001:** Basic information in residential construction.

Traditional Houses in Southern Shaanxi	Data Information
Basic Information	The number of rooms is 5. The building area is 12.6 m^2^ for the bedroom, 12.6 m^2^ for the kitchen, and 17.64 m^2^ for the living room.
Roof	8 mm blue brick + 3 mm waterproof coiled material + 15 mm cement mortar + 3 mm lime mortar
Wall	3 mm lime mortar + 3 mm cement mortar + 240 mm clay brick masonry + 3 mm cement mortar + 3 mm lime mortar
Window	Wooden windows are made of 6 mm single-layer colorless transparent glass, the heat transfer coefficient is 4.7 W/(m^2^·K), the shading coefficient is 0.8, the visible light transmittance is 0.77, the air permeability is 1.0 m^3^/(m·h).
Door	50 mm wood
Floor	120 mm gravel, pebble concrete + 10 mm cement mortar

**Table 2 ijerph-18-03745-t002:** Properties of residential materials.

Material	Density (Kg/m^3^)	Thermal Conductivity (W/m·K)	Specific Heat Capacity (J/Kg·K)	Thermal Storage Coefficient W/(m^2^·K)	Vapor Permeability Coefficient g/(m·h·kPa)
Cement mortar	1800	0.93	1050	11.37	0.021
Clay brick masonry	1700	0.76	1050	9.933	0
Lime mortar	1600	0.81	1050	10.07	0.0443
Crushed stone, pebble concrete	2300	1.51	920	15.36	0.0173
Blue brick	2000	1.16	920	12.56	0
Waterproof materials	600	0.17	1470	3.302	0
Wood	500	0.14	2510	3.575	0
Aluminum	2700	203	920	191.495	0

Note: The data come from the thermophysical properties and data manual of building materials.

## Data Availability

The datasets used and analyzed during the current study are available from the corresponding author on request.

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
