# Peer review of "Study on the Passive Heating System of a Heated Cooking Wall in Dwellings: A Case Study of Traditional Dwellings in Southern Shaanxi, China"

_ijerph, 2021, doi:10.3390/ijerph18073745_

Round 1

Reviewer 1 Report

The authors have carried out an interesting investigation on passive heating system of cooking heated wall in dwellings of Southern Shaanxi, China. However, the text of the manuscript has not been carefully performed and its quality could be significantly improved. Repetitions, disordered ideas and  spelling errors exist. The authors should write it in a more fluid and orderly way. An English review is necessary as well as a restructuring of the manuscript.

Author Response

Dear editor and reviewer 1:

  First of all, thanks to the editor for promptly notifying us that there is information about the paper. We also thank the reviewers for the time and effort that they have put into reviewing the previous version of the manuscript, and their suggestions have enabled us to improve our work.

  In addition, according to the instructions provided in your letter, we uploaded the revised manuscript using the revision mode in Microsoft Word, with all the changes highlighted in red. This letter is our point-by-point response to the comments made by the reviewers. These comments were copied and marked in black, and then we directly responded in red.

  Lastly, we would like to express our sincere thanks again to the reviewers for their constructive and positive comments!

With best wishes,

Simin Yang, Dewancker Bart, Shuo Chen.

Reviewer 2 Report

The manuscript is interesting and well written. The authors are urged to consider the following remarks in a revised submission of the manuscript:

  1. The resolution of Figs. 1d needs to be improved, because the text is quite blurry.
  2. The heading of Paragraph 2.1.2 should be “Case Study” instead of “Study Case”.
  3. In Paragraph 2.1.2, the methodology is described as instructions to a person. Throughout the article, text should be in third person, unless it is a direct quote (e.g. from an interview). Please revise accordingly.
  4. Overall, the description of methodology needs to be improved. For example, one of the steps is “Problem identification”. There, as the methodology it is stated that “Through the analysis of the data, it is found that the passive 244 heating of dwellings needs to be improved”, which is a result, not a method.
  5. Also, the methodology should be separated from the case study. A section is typically dedicated to the description of methodology.
  6. Field research was conducted. Are there photos from that? If so, it would be of interest to the readers to see some indicative ones.
  7. The resolution of images in Fig. 2 needs to be improved.
  8. The resolution of images in Fig. 3 needs to be improved.
  9. Why are the descriptions and the scales in Fig. 4 rotated? Please amend appropriately, as this makes it difficult for the reader to visualise.
  10. The formulas need to be typed in the appropriate format.
  11. The key to the formulas needs to be better expressed. E.g. after Formula 1, it is “Formula I is turbulence intensity and re is Reynolds number”, while it should be “where: I is the turbulence intensity, and re is the Reynolds number”.
  12. In Sub-section 2.3 it is mentioned that “The software used in this simulation is Ansys, Openstudio, Ladybug, Energyplus, etc.”. However, no other software package is mentioned apart from the ones already there. Please amend accordingly.
  13. The descriptions and the scales in Figs. 6, 7, 8, 9, and 11 are also rotated. Please amend appropriately, as this makes it difficult for the reader to visualise.
  14. Also, it is good practice that the authors want to keep consistency in the units and provide the equivalent temperatures in degrees Celsius, but it would be preferable to just change the values on the scale instead.
  15. Please note that the next figure after Fig. 9 is Fig. 11. As there is a second Fig. 10 after that, it is clearly a typographical mistake. Please amend accordingly.

Author Response

Dear editor and reviewer 2:

  First of all, thanks to the editor for promptly notifying us that there is information about the paper. We also thank the reviewers for the time and effort that they have put into reviewing the previous version of the manuscript, and their suggestions have enabled us to improve our work.

  In addition, according to the instructions provided in your letter, we uploaded the revised manuscript using the revision mode in Microsoft Word, with all the changes highlighted in red. This letter is our point-by-point response to the comments made by the reviewers. These comments were copied and marked in black, and then we directly responded in red.

  Lastly, we would like to express our sincere thanks again to the reviewers for their constructive and positive comments!

With best wishes,

Simin Yang, Dewancker Bart, Shuo Chen.

Round 2

Reviewer 1 Report

The reviewer appreciates the huge effort of authors to accomplish the reviewer’s comments and suggestions for a revision of the original submission.

Nevertheless, after all these substantial modifications, I have detected some minor editing errors in the text and in the references sections.

Author Response

(The authors gave the same response as above.)

Reviewer 2 Report

The authors have put an effort to address all reviewers’ remarks, which is evident in the revised submission. They are urged to consider the following remarks on the revised manuscript:

Major remarks:

  1. The research methodology section is still just a paragraph. It needs to be discussed in detail, so that the reader can understand exactly how all results were yielded without additional effort.

Minor remarks:

  1. The caption of Fig. 8 is too large. As the specific times are shown under each figure, an asterisk (*) could be added after the time at the images that are ‘without solar radiation’ and the caption could be:

Figure 8. Natural heating at defined times; * refers to heating without solar radiation.

  1. Similarly, for Fig. 10, it could be:

Figure 10. Wall heating for specific times.

  1. Similarly, for Fig. 11:

Figure 11. Cooking and heating at defined times; * refers to heating without solar radiation.

  1. Similarly, for Fig. 12:

Figure 12. Temperature after the heating system was powered off; * refers to heating without solar radiation.

Author Response

(The authors gave the same response as above.)
